# Physiological Mechanism through Which Al Toxicity Inhibits Peanut Root Growth

**DOI:** 10.3390/plants13020325

**Published:** 2024-01-22

**Authors:** Jianning Shi, Min Zhao, Feng Zhang, Didi Feng, Shaoxia Yang, Yingbin Xue, Ying Liu

**Affiliations:** 1Department of Biotechnology, College of Coastal Agricultural Sciences, Guangdong Ocean University, Zhanjiang 524088, China; 2Department of Agronomy, College of Coastal Agricultural Sciences, Guangdong Ocean University, Zhanjiang 524088, China

**Keywords:** peanut, Al toxicity, root growth, physiological mechanism

## Abstract

Al (Aluminum) poisoning is a significant limitation to crop yield in acid soil. However, the physiological process involved in the peanut root response to Al poisoning has not been clarified yet and requires further research. In order to investigate the influence of Al toxicity stress on peanut roots, this study employed various methods, including root phenotype analysis, scanning of the root, measuring the physical response indices of the root, measurement of the hormone level in the root, and quantitative PCR (qPCR). This research aimed to explore the physiological mechanism underlying the reaction of peanut roots to Al toxicity. The findings revealed that Al poisoning inhibits the development of peanut roots, resulting in reduced biomass, length, surface area, and volume. Al also significantly affects antioxidant oxidase activity and proline and malondialdehyde contents in peanut roots. Furthermore, Al toxicity led to increased accumulations of Al and Fe in peanut roots, while the contents of zinc (Zn), cuprum (Cu), manganese (Mn), kalium (K), magnesium (Mg), and calcium (Ca) decreased. The hormone content and related gene expression in peanut roots also exhibited significant changes. High concentrations of Al trigger cellular defense mechanisms, resulting in differentially expressed antioxidase genes and enhanced activity of antioxidases to eliminate excessive ROS (reactive oxygen species). Additionally, the differential expression of hormone-related genes in a high-Al environment affects plant hormones, ultimately leading to various negative effects, for example, decreased biomass of roots and hindered root development. The purpose of this study was to explore the physiological response mechanism of peanut roots subjected to aluminum toxicity stress, and the findings of this research will provide a basis for cultivating Al-resistant peanut varieties.

## 1. Introduction

With the continuous growth of industrial activities, the emission of acid gases has increased, leading to a more significant effect of acid deposition on the environment. Consequently, the prevalence and severity of acid deposition in certain regions have increased [1,2]. Global statistics indicate that about thirty percent of the land in the world consists of acidic soil [3]. Previous studies have demonstrated that soil acidity and high cation exchange capacity can elevate the soil aluminum (Al) concentration [4]. As soil acidification worsens, Al dissolves in the soil, causing Al toxicity to become a major factor restricting the growth of plants in acid soil [5].

Al is the largest quantity of metallic element found in the earthcrust [6]. In acidic soil, the different forms of Al that affect plant growth include aluminum ions (Al^3+^), Al(OH)_2_^+^, Al(OH)_4_^−^, and Al(OH)^2+^ [5,7]. Excessive Al can hinder plant growth in multiple ways. First, it restrains the growth of the root and hinders the absorption of water and nutrients [8]. For instance, high-Al treatment reduces the levels of P, Ca, N, Mg, K, Fe, and other elements in the ground and roots of *Brassica napus* [8]. Second, exposure to Al toxicity stress increases the content of Al^3+^ in plant cell walls and the cytoplasm, leading to the suppression of cell division and elongation [9]. Studies have shown that rice (*Oryza sativa*) cells exhibit various chromosomal abnormalities and abnormal cell division when exposed to Al toxicity, including chromosome adhesion, hysteresis, micronuclei, binuclei, and multinuclei cells [9]. Third, Al toxicity stress can damage the photosynthetic system and reduce pigment content, resulting in decreased photosynthesis in leaves [10]. Research has demonstrated that, compared with those of other organelles, Al-sensitive rye (*Secale cereale*) exhibit enlarged chloroplasts when subjected to Al toxicity stress, with increased Al accumulation. This leads to a decrease in chlorophyll content and an impairment of photosynthetic function [10]. Additionally, excessive accumulation of Al can lead to an increase in ROS (reactive oxygen species), which results in the peroxidation of lipids and dysfunction of organelles in the plasma membrane [11]. Consequently, plant roots produce major antioxidases, such as CAT (catalase), SOD (superoxide dismutase), and POD (peroxidase), to counteract the increase in ROS and free radicals caused by Al toxicity-related stress [11]. For example, the liveness of CAT, POD, SOD, and APX (ascorbate peroxidase) is significantly increased in the root tip cells of the Al-resistant Chinese fir (*Cunninghamia lanceolata*) variety YX01 after exposure to Al stress [11]. Research shows that the ROS burst induced by Al stress triggers mitochondria-dependent programmed cell death (PCD) in peanut root tip cells [7]. Other studies have shown that the quantity of fruit, mass of seeds, and yield of Al-sensitive varieties of *Brassica napus* are significantly lower than those of Al-resistant varieties under Al poisoning [12]. Therefore, when crops are exposed to Al toxicity stress, it severely affects their normal growth and reduces crop yield [10,11,12,13].

Peanut (*Arachis hypogaea*) is a significant source crop with high levels of fat and protein [14]. Its production is crucial for increasing the supply of oil and protein, thereby promoting agricultural production [15]. Currently, China has a peanut cultivation region of approximately 4.8 million hectares, with a yield of approximately 18 million tons. This has greatly contributed to China’s oil production and ensured an effective supply of vegetable oil [16]. Studies have shown that Al toxicity has a significant role in restricting peanut development on acidic soil [14]. However, there is a lack of research on peanut roots subjected to Al toxicity stress, particularly regarding the physiological response mechanism of peanut roots to Al toxicity stress.

Phytohormones play an important role in adjusting the growth of plants and helping roots adapt to stress [17]. Certain exogenously added regulatory factors (EARF), such as nutrient elements and phytohormones, play an important role in lowering Al toxicity in plants [13]. Phytohormones such as ABA (abscisic acid), JA (jasmonic acid), SA (salicylic acid), and GA (gibberellinic acid) have effects on cell elongation, differentiation, and proliferation in the roots of plants [18]. Additionally, JA participates in the formation and growth of plant root hairs [19], while high levels of SA can hinder the growth of taproots and lateral roots [19,20]. Moreover, ABA inhibits cell division and differentiation of stem cells, thereby restraining the growth of primary roots in plants [21,22]. GA regulates the area size of the root meristem in plants and is particularly important for GA biosynthesis in the cortex and inner cortex [23,24]. GA at the root tip stimulates the development and proliferation of root cells [24,25]. Despite the key role of plant hormones in the growth of roots, the regulatory mechanisms of plant hormones on peanut root growth under Al toxicity stress have not been investigated.

The peanut roots are essential for getting nutrients [26]. However, the physiological response mechanism of peanut roots to Al toxicity stress has not yet been established. However, the physiological regulatory pathways through which peanut roots respond to Al toxicity stress still need to be explored. The goal of this study was to investigate the physiological response mechanism of peanut roots to aluminum toxicity stress, and the findings would serve as a foundation for the development of novel Al-resistant peanut types.

## 2. Results

### 2.1. Effects of Al Poisoning on the Biomass of Roots in Peanut Plants

To analyze the influences of Al toxicity stress on the biomass of peanut roots, changes in the dry weight and fresh weight of peanut roots were measured under varying concentrations of Al. The results showed significant inhibition of peanut root biomass under Al toxicity stress (Figure 1A–F and Figure 2A,B). Initially, the peanut root biomass decreased as the concentration of Al increased. However, once the Al toxicity stress reached a certain level, there was no longer a significant decrease in peanut root biomass at the different Al concentrations (Figure 1A–F and Figure 2A,B).

In the experimental setup, a control group was treated with 0 mmol/L Al ions, while exogenous Al was applied at concentrations of 0.5, 1.0, 2.0, 4.0, and 8.0 mmol/L for a duration of 20 days. In comparison to the control group, the fresh weight of the peanut roots in the treatment groups markedly decreased by 43.05%, 51.43%, 53.37%, 49.86%, and 55.49%, respectively, with increasing concentrations of Al (Figure 2A).

Furthermore, in comparison to those in the control group, the dry weight of peanut roots treated with 0.5 mmol/L Al decreased by 10.25%. Although this decrease was not statistically significant, higher concentrations of Al (1.0, 2.0, 4.0, and 8.0 mmol/L) caused significant decreases in peanut root dry weight—33.85%, 35.40%, 31.37%, and 43.48%, respectively (Figure 2B).

### 2.2. Influences of Al Poisoning on Development of Peanut Roots

For analyzing the influences of Al toxicity stress on the development of peanut roots, the changes in peanut root growth under various concentrations of Al were measured. Exposure to Al toxicity had a certain effect on peanut root growth and development (Figure 3A–E). Exogenous Al was administered at concentrations of 0, 0.5, 1.0, 2.0, 4.0, and 8.0 mmol/L for a duration of 20 days. Comparatively, the control group (0 mmol/L Al) exhibited significant decreases in total root length of 51.63%, 61.75%, 67.41%, 69.96%, and 75.94%, respectively (Figure 3A). Furthermore, the average root diameter significantly increased by 38.30%, 32.02%, 37.66%, 39.77%, and 58.36%, respectively (Figure 3B). The total root volume also significantly decreased by 37.98%, 47.96%, 45.33%, 48.80%, and 48.51%, respectively (Figure 3C). Additionally, the total root surface area decreased considerably, by 44.89%, 51.96%, 64.60%, 63.91%, and 66.66% (Figure 3D), while the total number of root tips also significantly decreased by 53.62%, 66.43%, 70.77%, 70.83%, and 79.12%, respectively (Figure 3E).

### 2.3. Results of Al Poisoning on the Physical Response Indices of Peanut Roots

To analyze the influences of Al poisoning on the physical response of peanut roots, this study evaluated various physiological response indices of peanut roots under different concentrations of Al. The findings revealed that the physical response indexes of peanut roots were significantly influenced by exposure to Al toxicity stress. Using 0 mmol/L Al as the control, the application of exogenous Al at concentrations of 0.5, 1.0, 2.0, 4.0, and 8.0 mmol/L for a period of 20 days had a notable effect on the physiological response of peanut roots. In comparison with that in the control group, the SOD liveness in the 0.5 and 8.0 mmol/L Al-treated groups was not markedly different. However, the SOD activity in the 1.0, 2.0, and 4.0 mmol/L Al-treated groups improved significantly, by 172.50%, 231.25%, and 351.25%, respectively (Figure 4A).

The POD activity in the 8.0 mmol/L Al-treated group reduced markedly by 58.12%, whereas the POD activity in the 0.5, 1.0, 2.0, and 4.0 mmol/L Al-treated groups increased significantly by 61.41%, 88.94%, 77.65%, and 48.71%, respectively (Figure 4B). No significant change in CAT activity was observed in the 8.0 mmol/L Al-treated group, whereas CAT activity in the 0.5, 1.0, 2.0, and 4.0 mmol/L Al-treated groups improved significantly by 83.78%, 129.73%, 175.68%, and 189.19%, respectively (Figure 4C). The APX activity in the 8.0 mmol/L treatment group showed no significant change, while the APX activity in the 0.5, 1.0, 2.0, and 4.0 mmol/L Al-treated groups elevated significantly by 42.19%, 57.81%, 135.94%, and 103.13%, respectively (Figure 4D).

Regarding biochemical parameters, the soluble protein content in the 8.0 mmol/L treatment group remained relatively constant, but in the 0.5, 1.0, 2.0, and 4.0 mmol/L Al-treated groups, it elevated significantly by 21.35%, 33.98%, 42.44%, and 60.80%, respectively (Figure 4E). The soluble sugar content in the 0.5 mmol/L Al-treated group did not obviously change, while in the 1.0, 2.0, 4.0, and 8.0 mmol/L Al-treated groups, it enhanced significantly by 46.96%, 72.22%, 99.25%, and 600.66%, respectively (Figure 4F). The proline content in the 0.5 and 8.0 mmol/L Al-treated groups exhibited no significant change, whereas in the 1.0, 2.0, and 4.0 mmol/L Al-treated groups, it increased markedly by 25.77%, 64.43%, and 107.26%, respectively (Figure 4G). The malondialdehyde (MDA) content in the 8.0 mmol/L Al-treated group reduced observably by 42.17%, and no obvious change was surveyed in the content of MDA in the 0.5 and 1.0 mmol/L Al-treated groups (Figure 4H). However, the content of MDA in the 2.0 and 4.0 mmol/L Al-treated groups enhanced significantly, by 40.26% and 86.13%, respectively (Figure 4H).

### 2.4. Influences of Al Poisoning on the Accumulation of Several Elements in Peanut Roots

The ion contents of ten different elements in peanut roots with varying Al concentrations were measured to analyze the effect of aluminum toxicity stress on ion accumulation. The control group received a 0 mmol/L aluminum treatment, while the samples were taken and dried after a 20-day treatment with 4.0 mmol/L exogenous aluminum. The contents of Al, K, Fe, Na, Mn, Cu, Mg, Zn, Se, and Ca were determined in peanut roots. In comparison with that in the control group, the content of Na in the Al-treated group was significantly elevated by 30.54% (Figure 5C). The contents of Al and Fe increased markedly by 18,840.51% and 127.13%, respectively (Figure 5A,G). Mg, K, Ca, Mn, Cu, Zn, and Se decreased significantly, by 79.42%, 25.23%, 74.62%, 80.55%, 48.78%, 13.46%, and 55.33%, respectively (Figure 5B,D–F,H–J).

### 2.5. Effects of Al Toxicity Stress on the Contents of Several Hormones in Peanut Roots

The contents of six different hormones in peanut roots were detected to investigate the influence of aluminum toxicity stress on hormone synthesis. The samples were taken under normal aluminum conditions at 4 mmol/L of Al. In comparison with those in the control group, the contents of ABA, SA, and gibberellin 3 (GA_3_) in the treatment groups significantly increased by 322.67%, 329.20%, and 240.00%, respectively (Figure 6A,D,F). However, zeatin, indole acetic acid (IAA), and JA significantly decreased by 70.00%, 63.80%, and 83.86%, respectively (Figure 6B,C,E).

### 2.6. Effects of Al Toxicity Stress on the Relative Transcript Levels of Several Types of Genes in Roots of Peanut

To analyze the influences of Al toxicity on relative expression of the genes in connection with physiological responses (*SOD1*, *POD1*, *CAT1*, *APX1*, and *APX2*; hormone synthesis (*ABA1*, *IAA1*, *Zeatin1*, *Zeatin2*, *SA2*, and *GA3.1*); and ion transport (*Nramp5*, *Nramp3.1*, *MTP1*, *MTP10*, and *VIT1*), we analyzed the expression of 15 genes under Al toxicity stress. The results demonstrated that, in comparison with those in the controls, the genes whose relative expression decreased significantly included *SOD1* (Figure 7A), *POD1* (Figure 7B), *CAT1* (Figure 7C), *APX1* (Figure 7D), *APX2* (Figure 7E), *ABA1* (Figure 7F), *Zeatin1* (Figure 7H), *Zeatin2* (Figure 7I), *SA2* (Figure 7J), *GA3.1* (Figure 7K), *Nramp5* (Figure 7L), *MTP10* (Figure 7O), and *VIT1* (Figure 7P). The genes whose relative expression increased significantly were *IAA1* (Figure 7G), *Nramp3.1* (Figure 7M), and *MTP1* (Figure 7N).

## 3. Discussion

In acidic soils, active Al ions are the primary factor that limits plant growth [27]. The main effectiveness of Al poisoning on plants is the suppression of root growth [28]. Our study demonstrated significant reductions in the fresh weight, dry weight, length of total root, surface area of total root, surface area of total root, and root tip number in the roots of peanuts subjected to Al poisoning, indicating severe suppression of the elongation of roots. Interestingly, the average diameter of the peanut roots increased significantly following exposure to Al toxicity. This may be caused by the close combining of Al with the cell wall of root cells in plants, which alters the structure and flexibility of the cell wall, reducing its ductility and disrupting the normal function of the plant cytoskeleton. Consequently, the root tip expands [29]. Similar findings have been reported in *Eucalyptus* clones, in which Al toxicity stress led to taproot enlargement and root tip shortening and thickening [30]. These changes may be due to alterations in the cell wall cellulose content under Al stress, which causes cell deformation and a shift in growth direction from vertical to horizontal, resulting in the expansion and thickening of the root tip base [30].

Under conditions of severe aluminum toxicity stress, there can be an imbalance between the generation and elimination of free radicals and ROS, leading to an oxidative stress response, enzyme inactivation, and peroxidation of cell membranes [31]. The antioxidant defense system and osmoregulation system play significant roles in plants’ ability to resist oxidative stress [32]. Plants produce various enzymes, including SOD, POD, CAT, and APX, which help them cope with oxidative stress [32]. Additionally, components such as soluble proteins, proline, and soluble sugars can help reduce the effects of Al poisoning [32]. The content of MDA is an essential indicator of lipid peroxidation in plant cell membranes [33]. Hence, the measurement of antioxidase activity and osmotic adjustment components is crucial for reflecting the metabolism activity and health level of cells [34].

The SOD family genes found in plants are categorized into three groups based on their metal cofactors: Cu/ZnSOD, MnSOD, and FeSOD [35]. SOD plays a crucial role as the first line of defense in plant antioxidant systems. In *Brassica napus* L., the genes *BnCuSOD1*, *BnCuSOD3*, *BnCuSOD14*, *BnFeSOD4*, *BnFeSOD5*, *BnFeSOD6*, *BnMnSOD2*, and *BnMnSOD10* were observed to be significantly upregulated under various abiotic stress conditions, such as cold, salt, drought, and flood [36]. In the present study, the differential expression of peanut *SOD1* and the activity of SOD in peanut roots may be influenced by variations in the absorption of Fe, Cu, Mn, and Zn affected by the presence of Al^3+^. The POD gene family refers to the class III peroxidase gene family, which is responsible for directing the synthesis of plant-specific REDOX enzymes in grapevine (*Vitis vinifera*) [37]. Previous research has demonstrated that treatment with H_2_O_2_ induces the upregulation of *MePOD13*, *MePOD17*, *MePOD23*, and *MePOD85* expression in *Manihot esculenta* [38]. PODs not only help remove ROS but also contribute to the accumulation of lignin in cell walls, triggering significant thickening and hardening of the cell wall, as observed in watermelon (*Citrullus lanatus*) [39]. The genes *ClPOD51* and *ClPOD54* are believed to play key roles in lignin synthesis in watermelon rinds [39]. In the present study, Al poisoning induced differential expression of *POD1*, which may not only impact the antioxidant capacity of peanuts but also influence root growth inhibition. The expression of *CAT* in certain plants is influenced by plant hormones [40]. For example, the expression of *AtCAT1* in *Arabidopsis thaliana* is induced by ABA [40]. In *Brassica napus*, the genes *BnCAT1*, *BnCAT3*, *BnCAT11*, and *BnCAT13* were significantly upregulated under low temperature, salt, ABA, and GA treatments [41]. Previous reports identified a total of 166 members of the peanut APX gene family, with *AhAPX4*, *AhAPX17*, and *AhAPX19* showing significant upregulation under abiotic stress (drought and cold) and treatment with plant hormones such as ABA and SA, indicating that the synthesis of APXs is induced by abiotic stress and that their differential expression is associated with hormone levels in peanut cells [42]. Thus, the changes in CAT and APX activities observed in this study are likely related not only to the oxidative damage caused by Al poisoning but also to alterations in hormone levels in peanut root cells.

In this study, there was a clear pattern observed in the liveness of SOD, CAT, POD, and APX, with a beginning enhancing followed by a reducing. When the concentration of Al was low, the liveness of those antioxidases was elevated, indicating that the peanut cell membrane system experienced some damage due to Al toxicity. POD, SOD, APX, and CAT are known to play a significant role in protecting the plant cell membrane system during aluminum poisoning [43]. However, as the concentration of Al increased to a certain extent, the activities of these enzymes started to decline. This decline might be attributed to the depletion of SOD, POD, CAT, and APX due to their involvement in protecting the peanut membrane system [44]. Alternatively, the decline could be a result of an excessively high Al concentration, which seriously damages the oxidation defense system of the peanut plants, as evidenced by the reduced activities of the enzymes [45]. Similar decreases in physiological response indices have also been observed under other abiotic stress conditions, such as in soybean (*Glycine max*) treated with microplastics [45] and in broad bean (*Vicia faba*) treated with lead [46].

The experiment also revealed trends in the levels of proline, soluble sugar, and soluble protein. The soluble protein and proline contents initially increased, followed by a decrease, while the soluble sugar content continuously increased. Notably, at the highest concentration tested, compared to those at the previous concentration, the soluble protein and proline levels decreased, while the soluble sugar level significantly increased. This finding suggests that peanuts respond synergistically to Al toxicity stress by adjusting the contents of osmotic adjustment components such as soluble protein, proline, and soluble sugar [43]. MDA, a byproduct of lipid peroxidation caused by oxidative damage, reflects the extent of peroxidation damage to the cytomembrane [47]. In this study, the MDA content exhibited an initial enhancement followed by a reduction. As the concentration of Al increased, additional peroxidative damage occurred in the peanuts due to Al toxicity, leading to a gradual increase in the MDA content. However, at the highest Al concentration, the MDA content decreased significantly compared to that at the previous concentration. This suggested that at higher Al concentrations, the antioxidant defense system in peanut root cells was severely damaged and possibly dysfunctional.

Based on these results, it can be inferred that under lower concentrations of Al toxicity stress, the antioxidant defense system and osmotic regulation system of peanuts can protect peanut root cells and minimize oxidative damage. However, under higher concentrations of Al toxicity stress, the antioxidant defense system of peanuts is severely impaired, while the osmotic regulation system continues to play a role.

In comparison with that in the control group, the accumulation of Al in peanut roots significantly increased after exposure to the Al treatment. This increase could be the cause of Al poisoning in peanut roots. Previous studies suggest that Al ions have a greater affinity for unmethylated pectin in the cytoderm, leading to a substantial buildup of Al in the root cell wall of plants. This accumulation results in increased rigidity in the cell wall and abnormal cytoskeletal structure [48]. Additionally, excessive Al in the cytoplasm can cause various chromosomal abnormalities, including chromosome adhesion, hysteresis, micronucleus formation, binucleus, and multinucleus cells. Consequently, these abnormalities can affect the division and differentiation of plant cells [9].

Notably, in this research, the contents of Mg, Ca, and Mn in the roots of peanuts decreased markedly. Research findings indicate an antagonistic relationship between Mg and Al in plants [49]. There is also competition between Mg and Al for binding sites on the cell wall and membrane. Inside the cell, Mg and Al compete for binding sites on oxygen-contributing compounds involved in energy metabolism, protein synthesis, and photosynthesis [49]. Similarly, Mn and Al also exhibit antagonistic effects [50]. When combined, the Al and Mn treatments mitigated the decrease in plant biomass caused by Al or Mn alone. This suggests that the two ions have antagonistic interactions that affect plant growth [51]. Ca is another component of pectin in plant cell walls and influences the fluidity of cell membranes [52]. Calcium ions serve as signaling molecules for physiological regulation, and they can compete with Al ions to bind to sites in pectin as cations [53]. In the present study, the decreases in the contents of Mg, Ca, and Mn in peanut roots may be attributed to the high concentration of Al ions occupying a significant number of binding sites in peanut root cells.

Furthermore, an increase in the Fe content was detected in peanut roots subjected to Al toxicity stress. Similar findings have been observed in soybeans exposed to Mn poisoning [54]. It is possible that the accumulation of Al in peanut roots also leads to an increase in Fe content. There may be a certain synergistic accumulation of these two ions in peanuts, and an increase in Fe content may contribute to improving peanut resistance to Al toxicity.

The NRAMP (natural resistance-associated macrophage protein) family plays a vital role in metal absorption and transportation in plants, with most *AhNRAMPs* being primarily expressed in androecia, roots, and seeds [55]. *NRAMPs* are crucial for the transport of metal ions across cell membranes. For instance, the expression levels of *GmNRAMP5a* and *GmNRAMP1a* are upregulated under Cu toxicity stress in *Glycine max* [56]. Metal-tolerance proteins (MTPs) also play a significant role in the transport of bivalent cations in plants [57]. For example, the expression levels of *GmMTP1.1* in soybean leaves and *GmMTP4.3* in roots significantly increased under treatment with bivalent Cd, Fe, Co, Zn, and Mn [58]. Previous studies have focused mainly on the involvement of the NRAMP and MTP families in the induction of bivalent metal ions, such as Fe^2+^ and Cd^2+^ [55,57]. In the present study, certain members of the peanut NRAMP and MTP families (*AhNramp5*, *AhNramp3.1*, *AhMTP1*, and *AhMTP10*) exhibited differential expression in peanut roots exposed to aluminum poisoning. These findings suggested that the peanut NRAMP and MTP families probably also have a critical role in peanut resistance to Al poisoning. Moreover, transporters associated with bivalent metal ions may not only participate in the transport of Mg, Fe, Ca, Mn, Cu, and Zn but also contribute to the transport of trivalent Al^3+^.

ABA is a hormone that is crucial for plant resistance to metal poisoning [59]. In the root system, ABA functions primarily in the meristem and elongation regions [59]. Abscisic aldehyde oxidase, 9-cis-epoxycarotenoid dioxygenase, and alcohol dehydrogenase (ABA2) impact ABA synthesis in plants in response to abiotic stress [59,60,61]. Research has revealed that the NGATHA1 transcription factor in *Arabidopsis thaliana* positively regulates ABA production during abiotic stress by activating *NCED3*, which encodes a key ABA biosynthesis agent [62]. Under specific concentrations of metal ions, ABA promotes primary root growth while inhibiting lateral root growth [63,64]. Plants can regulate ABA content under abiotic stress by modulating the transcription levels of several genes associated with ABA biosynthesis [59]. Metal stress, such as cadmium, nickel, zinc, and Al exposure, impacts the ABA content in plants [61]. In the present study, significant increases in ABA content were observed when peanut roots suffered from Al poisoning. This difference may be attributed to the induction of genes in connection with ABA biosynthesis in peanut roots by Al, suggesting that peanuts enhance the ABA content to counter Al toxicity stress.

Zeatin is typically produced by rapidly dividing cells, such as the root tip and bud, and plays a vital role in promoting plant cell division, root extension, leaf expansion, and seed germination [65,66]. Isopentenyltransferases (IPTs) are pivotal enzymes that catalyze cytokinin biosynthesis, while IAA influences Zeatin biosynthesis by regulating *IPT* expression [67]. Studies have demonstrated that Arabidopsis *AtIPT9* stimulates cytokinin synthesis and enhances root growth [68]. In the current study, the zeatin level in the roots of peanuts was significantly reduced under Al poisoning. It is possible that the induction of Al ions influenced the expression of relevant genes, hindered Zeatin synthesis, and inhibited cell division and root growth in peanut roots.

IAA is involved in regulating all aspects of normal plant growth and development, including synthesis, metabolism, polar transport, and self-signal transduction [69]. The stability of IAA levels also represents the coping strategies of plants for changes in the environment [70,71]. The auxin-related genes such as *PINFORMED* (*PIN*) and *AUXIN1/LIKE AUX1* (*AUX1/LAX*) involve adjusting the polar transportation of growth hormones, thus further controlling and optimizing the growth hormone distribution in root tips [72,73]. The metabolism and dynamic polarity distribution of IAA are influenced by metal ions [69]. In this study, the IAA level in peanut roots under Al toxicity stress was lower than that under normal conditions, indicating that Al toxicity inhibited peanut root growth.

SA regulates biological and abiotic stress responses in plants, promotes ROS production through antioxidant defense systems, and induces gene expression [74]. Under stress, a low SA concentration can facilitate the growth of plants, while a high SA concentration can inhibit plant growth [75,76]. This is because a high SA concentration affects the gene expression of *PIN2*, which acts as a vital gene for adjusting the synthesis of hormone, and downregulates its expression, thereby reducing hormone production in the root tip meristem and conclusively restraining the growth of roots and the development of root hair in plants [75]. In this study, the SA level in peanut roots significantly increased under Al toxicity stress. It was possible that SA synthesis induced by Al toxicity stress led to an increase in the SA level in the roots of peanuts. As a result, the growth hormone content and distribution in peanut roots were affected, leading to peanut root dysplasia.

When plants experience stress, JA can assist in their adaptation to stress [77]. In the case of blueberries (*Vaccinium corymbosum*) exposed to Al toxicity, the application of exogenous methyl jasmonate (MeJA) has been found to mitigate damage caused by Al toxicity stress [78]. There is an antagonistic relationship between JA and SA, where SA inhibits the expression of JA response genes, and JA may weaken the immune response induced by SA [79]. JA also participates in the gene expression of certain *metal ion transporters*, such as *FRO2,* and *IRT1*, which impact the accumulation of metal ions in roots [80]. In the present study, the JA level in the roots of peanuts was substantially decreased under Al toxicity stress. This reduction may be attributed to the antagonistic effect between JA and SA, as an increase in SA content results in a decrease in JA content.

GA_3_ plays a significant role in promoting plant germination, stem extension, leaf growth, flowering, and fruit development [81]. In Arabidopsis, *GA2-oxidase* (*GA2ox*), *GA3ox* (*GA3-oxidase*), and *GA20ox* (*GA 20-oxidase*) have been identified as key genes involved in the synthesis and degradation of GA_3_ [82]. An antagonistic relationship exists between GA_3_ and ABA, which regulates various plant development processes, including root growth [83]. Previous studies have demonstrated that exposure to Al leads to a reduction in GA_3_ content and an increase in ABA content in the roots of *Pinus massoniana* [84]. However, in the present study, the GA_3_ level in peanut roots increased with increasing ABA content under Al toxicity stress. These findings contradict previous research results and may be attributed to species-specific coping strategies [82]. It is possible that Al toxicity stress influences the expression of genes associated with GA_3_ synthesis in peanut roots, thereby enabling peanuts to increase their tolerance to Al toxicity stress by increasing the GA_3_ content in their roots.

In this study, the DEGs were categorized into three main groups: genes related to physiological indices (*SOD1*, *POD1*, *CAT1*, *APX1*, *and APX2*), hormone-related genes (*ABA1*, *IAA1*, *Zeatin1*, *Zeatin2*, *SA2*, and *GA3.1*), and ion transport genes (*Nramp5*, *Nramp3.1*, *MTP1*, *MTP10*, and *VIT1*). Physiological index response-related genes and hormone-related genes have crucial roles in adjusting plant responses to abiotic stress and recognizing ROS [85]. The differential expression of physiological index response-related genes and hormone-related genes in the presence of Al suggests that plants respond to the stress of Al toxicity by activating these genes to resist stress and eliminate excessive ROS in roots. The NRAMP family is highly important for metal absorption and transportation in plants, with most *AhNRAMPs* being primarily expressed in androecia, roots, and immature seeds [55]. MTPs are also vital for bivalent cation transport in plants [86]. Previous studies have linked *NRAMPs* and *MTPs* to the transportation of bivalent metal ions such as Fe^2+^ and Cd^2+^ [55,87]. In the present study, several peanut *AhNRAMPs* (*Nramp5* and *Nramp3.1*) and *AhMTPs* (*MTP1* and *MTP10*) exhibited differential expression in peanut roots exposed to Al toxicity, suggesting that the peanut NRAMP and MTP families may also play significant roles in peanut resistance to Al toxicity stress. Furthermore, transporters associated with bivalent metal ions might be involved in the transport of trivalent Al^3+^, providing new perspectives for studying peanut Al tolerance.

On account of the experimental results of this research, the physiological regulatory mechanism of peanut roots in responding to Al toxicity is summarized in Figure 8. First, under Al poisoning and high levels of exogenous Al, there was an increase in the absorption of Al by root cells. This excess Al led to intracellular oxidative damage (Figure 8I), while cells regulated their internal environment through osmotic substances (Figure 8II) to maintain stability. A high Al concentration activated cellular defense mechanisms, resulting in differential gene expression of antioxidase-encoding genes (Figure 8III). Consequently, the activities of antioxidases were enhanced (Figure 8VI), effectively eliminating excess ROS. Under high-Al conditions, phytohormone-relevant genes were expressed differentially (Figure 8IV), and ion transshipment-relevant genes were also expressed differentially (Figure 8V), causing changes in plant hormones (Figure 8VII) and a reduced absorption capacity for mineral nutrients (Figure 8VIII). Ultimately, this led to phenotypes such as decreased biomass of roots and inhibited growth of roots in peanut plants.

## 4. Materials and Methods

### 4.1. Plant Materials and Hydroponic Treatment

The plant material was the peanut variety Zhanyou 62, bred by the ZAAS (Zhanjiang Academy of Agricultural Sciences) in Zhanjiang, Guangdong Province, China. It was cultivated in the plant culture room of the College of Coastal Agricultural Sciences at Guangdong Ocean University (E: 110.30311, N: 21.15005). Peanut seeds with complete seed coats and uniform sizes were selected. The plants were sterilized with 1% NaClO (Solarbio, Beijing, China) and planted in silica sand. The peanut seeds were buried and cultured in sand for 10 days. Afterward, the uniformly grown peanut plants were transferred to a 15 L plastic box and provided with an improved Hoagland nutrient solution [87] for hydroponic cultivation. The hydroponic nutrient solution contained 25 µmol/L MgCl_2_, 400 µmol/L NH_4_NO_3_, 1500 µmol/L KNO_3_, 40 µmol/L Fe-EDTA (Na), 1200 µmol/L Ca(NO_3_)_2_·4H_2_O, 500 µmol/L MgSO_4_·7H_2_O, 1.5 µmol/L ZnSO_4_·7H_2_O, 0.5 µmol/L CuSO_4_·5H_2_O, 300 µmol/L (NH_4_)_2_SO_4_, 300 µmol/L K_2_SO_4_, 1.5 µmol/L MnSO_4_·H_2_O, 0.16 µmol/L (NH4)_5_MoO_24_·4H_2_O, 2.5 µmol/L NaB_4_O_7_·10H_2_O, and 500 µmol/L KH_2_PO_4_. All chemical reagents used were of analytical grade (Kermel, Tianjin, China). Al_2_(SO_4_)_3_·18H_2_O (Shanghai Reagent, Shanghai, China) was used as the source of Al, and Al^3+^ was added to the nutrient solution at concentrations of 0, 0.5, 1.0, 2.0, 4.0, and 8.0 mmol/L for hydroponic treatment of peanuts. The control group was treated with an Al^3+^ concentration of 0 mmol/L. Each treatment was repeated three times. The peanut plants were cultivated under the following conditions: a temperature of 25–30 °C during the daytime and 18–22 °C at nighttime, with a light period of approximately 12 h/d and a light intensity of 2000 lux. The nutrient solutions were changed every 5 days. After 20 days of Al treatment, the peanut roots were collected.

### 4.2. Measurement of Dry and Fresh Weight of Roots in Peanut

The root fresh weight of each peanut plant was measured using an BS124S electronic balance (Sartorius, Göttingen, Germany). Then, the root samples were moved to an electric blast constant temperature oven (Yiheng, Shanghai, China) and dried at a constant temperature of 60 °C for one week [88]. The dry weight of each treated sample was measured for three biological replicates.

### 4.3. Determination of Peanut Root Morphological Indices

A WinRHIZO LA6400 XL root scanner (Regent, Vancouver, BC, Canada) was used to scan the morphological indices of the peanut roots and capture images of each treated peanut root system. WinRHIZO software (WinRHIZO 2013e Professional Edition) was subsequently used to analyze and obtain data concerning the surface area, volume, total length, and number of roots [89]. Three measurements were obtained for each treatment.

### 4.4. Determination of Peanut Root Physiological Response Indices

As mentioned previously, peanut roots were treated with Al_2_(SO_4_)_3_·18H_2_O as an aluminum source for 20 days using various concentrations of Al^3+^ (0, 0.5, 1.0, 2.0, 4.0, and 8 mmol/L). After the treated peanut roots were harvested, eight physiological response indices were measured. The proline level was determined by means of the sulfosalicylic acid method [90]. The level of soluble protein was tested by applying the Coomassie bright blue staining technology [85]. The soluble sugar level was tested by means of the anthrone-sulfuric acid colorimetry method [91]. The level of MDA was measured by applying the thiobarbituric acid method [92]. POD liveness was determined by means of the guaiacol way [93]. SOD liveness was tested by application of the nitrogen blue tetrazole method [94]. CAT life was tested by means of a spectrophotometer (Yuexi UV-5100B, Shanghai, China) [95]. The activity of APX was tested by applying the experimental technology demonstrated by Li et al. [96].

### 4.5. Determination of Ion Content in Peanut Roots

Peanut roots were treated with 0 (control group) or 4.0 (al poison treatment group) mmol/L Al^3+^. The dry root samples (0.15 g) were completely dissolved in nitric acid (Guangzhou Reagent, Guangzhou, China). The levels of Al, Mg, Na, K, Fe, Ca, Mn, Zn, Cu, and Se in peanut roots were tested using PS7800 ICP–AES (inductively coupled plasma atomic emission spectrometry) (Hitachi, Tokyo, Japan) [86,97]. Each index was analyzed using three biological replicates.

### 4.6. Determination of the 6 Kinds of Hormones in Peanut Roots

Peanut roots were treated with 0 or 4.0 mmol/L Al^3+^ for 20 days of culture, after which endogenous hormones were extracted from the roots. The contents of endogenous hormones, including ABA, IAA, zeatin (ZT), SA, JA, and GA_3_, in peanut roots were measured using high-performance liquid chromatography (HPLC) (AGLIENT 1290, Santa Clara, CA, USA) combined with SCIEX-6500Qtrap MS/MS (tandem mass spectrometry) (AB, Madison, WI, USA). Internal reference materials were supplemented with the extract to correct the determination outcome [54,98]. The external standard substances used, such as zeatin, SA, ABA, JA, IAA, and GA_3_, were chromatography-pure reagents (Sigma, Saint Louis, MI, USA). The internal standard substances used were deuterated zeatin (D-zeatin), deuterated SA (D-SA), deuterated ABA (D-ABA), deuterated JA (D-JA), deuterated IAA (D-IAA), and deuterated GA_3_ (D-GA_3_) (Sigma, Saint Louis, MI, USA). C18 QuECherS (Amperex, Shanghai, China) was used as a filler for the column. The CH_3_OH and C_2_H_3_N used in the test were chromatography-pure (Merck, Darmstadt, Germany). Preparation of the operating fluid and calculation of the hormone standard curve, as well as extraction of hormones from peanut roots, were conducted following the methods reported in reference [85]. Appendix A present the HPLC gradient parameters, mass spectrometry parameters, and selected monitoring conditions for protonated or deprotonated plant hormone reactions, respectively.

### 4.7. Fluorescence Quantitative PCR (qPCR) Analysis

In accordance with the results of transcriptome sequencing in peanuts, several genes differentially expressed in response to metal ion stress and potentially linked to antioxidant and hormone synthesis were identified [88]. In this study, qPCR was adopted to analyze gene expression in response to aluminum toxicity stress in peanuts. Peanut roots were collected after 20 days of dealing with 0 or 4.0 mmol/L of Al. The RNA was isolated from the roots of peanuts via the Kit of MolPure Plant RNA (Yeasen, Shanghai, China). The cDNA was obtained through reverse transcription by means of the Kit of HifairII 1st Strand cDNA Synthesis (Yeasen, Shanghai, China). The expression of genes in the peanut roots was detected using Hieff UNICON Universal Blue qPCR SYBR Green Master Mix reagent (Yeasen, Shanghai, China) and Bio-Rad fluorescence quantitative PCR (CFX Connect Optics Module, Hercules, CA, USA) [99].

For qPCR, a 20 μL reaction mixture was utilized, consisting of 10 μL of Hieff UNICON Universal Blue qPCR SYBR Green Master Mix reagent, 1 μL of cDNA template, 0.5 μL of downstream primers, 0.5 μL of upstream primers, and 8 μL of nuclease-free water. The qPCR reaction procedures were performed as follows: 95 °C for 10 min; 95 °C for 15 s; 60 °C for 1 min; and 40 cycles [100]. *AhUbiquitin* (DQ887087.1) was selected as the internal control gene [101]. The gene expression levels were determined via the following formula: gene relative expression level = expression level of targeted gene/expression level of internal control gene [72]. The primers used for qPCR are displayed in Appendix A.

### 4.8. Data Analysis

Microsoft Excel 2010 (Microsoft, Redmond, WA, USA) and SPSS (Statistic Package for Social Science) software version 19.0 (IBM Corporation, New York, NY, USA) were applied to statistical analysis. A student’s t test was adopted for statistical comparison and significance analysis between the two groups of data. The Waller-Duncan test was used to compare the significance of differences across multiple groups of data [85].

## 5. Conclusions

Based on the aforementioned results, an analysis was conducted on the physiological regulatory mechanism of peanut roots in responding to Al poisoning. Initially, the entrance of Al into peanut root cells increased due to the high presence of exogenous Al during Al toxicity stress. Throughout this process, the increased levels of Al activated cellular defense mechanisms, thereby altering the expression of antioxidase-encoding genes and enhancing the activity of antioxidases. Consequently, the excess ROS were effectively eliminated. Additionally, exposure to Al ions during Al toxicity stress led to the induction of genes associated with plant hormones and metal ion transport. Consequently, this induction caused changes in the expression levels of plant hormones and metal ion transport-related genes, ultimately resulting in a range of symptoms, such as decreased biomass of roots, restrained growth of roots, and abnormal accumulation of metal ions in peanut roots. The consequences of this study establish a foundation for further research into the physiological and molecular mechanisms referred to in the peanut root responding to Al poisoning. Moreover, these findings offer a theoretical grounding for the development of new Al-resistant peanut varieties through breeding efforts.

## Figures and Tables

**Figure 1 plants-13-00325-f001:**
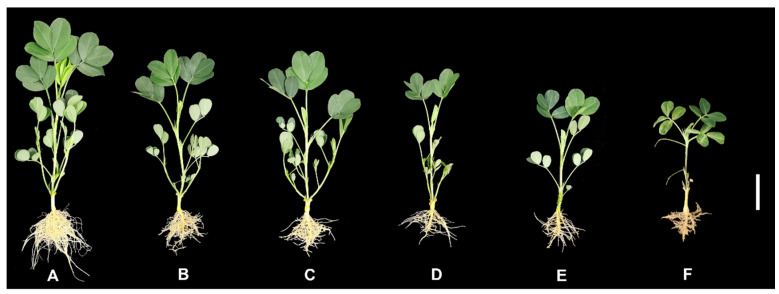
Phenotypes of peanuts after treatment with various Al concentrations. Peanuts were dealt with at various concentrations of Al for 20 days. (**A**) 0; (**B**) 0.5; (**C**) 1.0; (**D**) 2.0; (**E**) 4.0; and (**F**) 8 mmol/L (Bar = 5 cm).

**Figure 2 plants-13-00325-f002:**
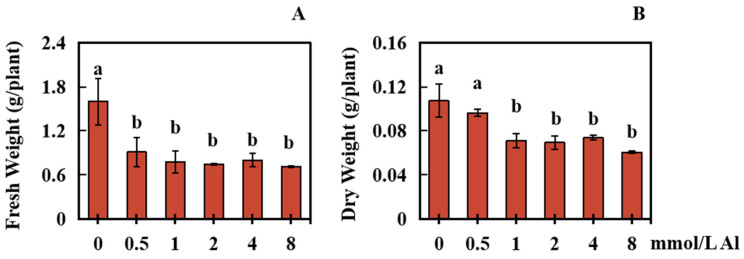
Effects of different Al concentrations on biomass of peanut roots. The fresh and dry weight data were collected after 20 days of treatment with exogenous Al at concentrations of 0.5, 1.0, 2.0, 4.0, and 8.0 mmol/L, with the control group being treated with 0 mmol/L Al. (**A**) Fresh root weight; (**B**) Dry root weight. The results were represented by the average value and SD (standard deviation) of three repeated biological experiments. The Waller–Duncan multiple comparison test and single-factor ANOVA were adopted to make a comparison for significant differences between the control group and treatment group with each concentration of Al toxicity. Different letters on the bar chart indicated significant differences between data (*p* < 0.05).

**Figure 3 plants-13-00325-f003:**
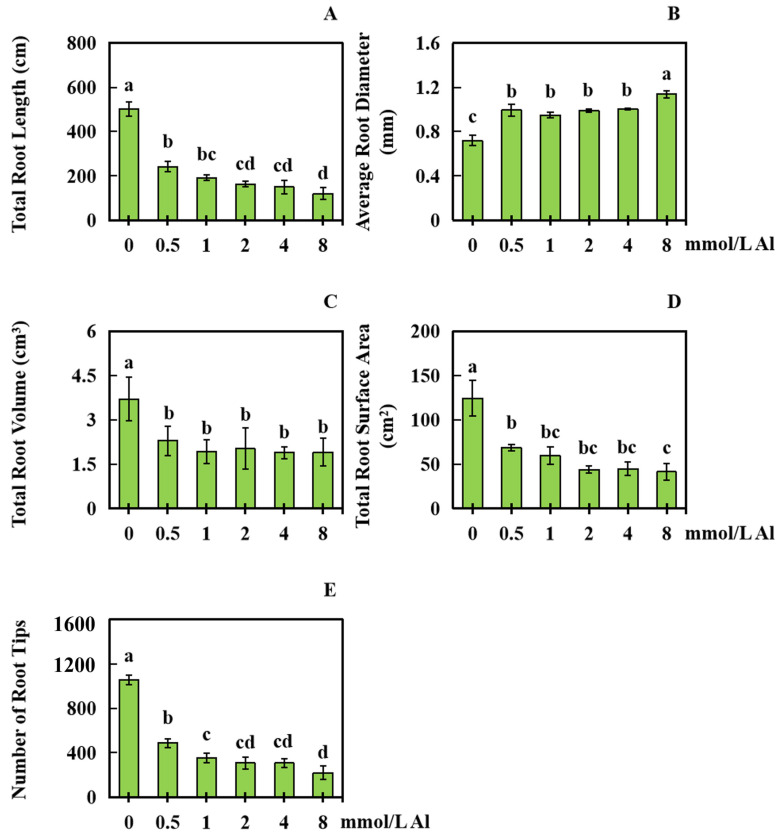
Peanut root development under different Al concentrations. With 0 mmol/L Al treatment as the control, root data were collected after 20 days of treatment with 0.5, 1.0, 2.0, 4.0, and 8.0 mmol/L exogenous Al. (**A**) Length of total root; (**B**) average diameter of root; (**C**) volume of total root; (**D**) surface area of total root; (**E**) root tip number. The results were represented by the average value and SD (standard deviation) of three repeated biological experiments. The Waller–Duncan multiple comparison test and single-factor ANOVA were adopted to make a comparison for significant differences between the control group and treatment group with each concentration of Al toxicity. Different letters on the bar chart indicated significant differences between data (*p* < 0.05).

**Figure 4 plants-13-00325-f004:**
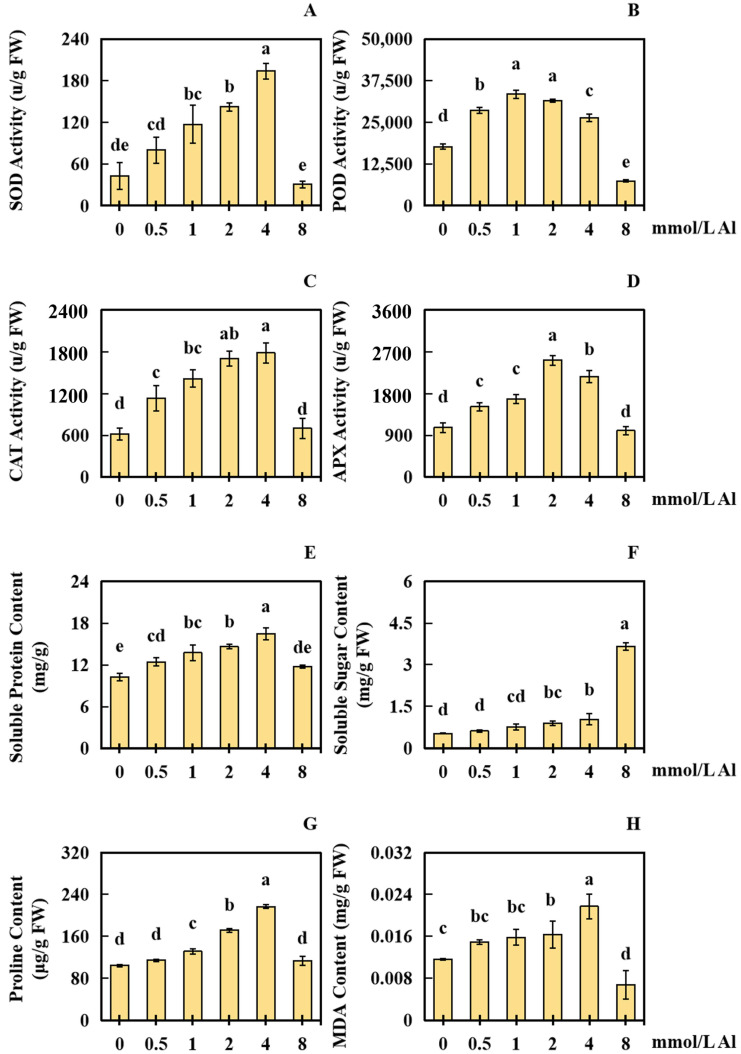
Results of the physiological response indices of peanut roots under aluminum toxicity stress. A 20-day treatment was conducted using exogenous aluminum concentrations of 0.5, 1.0, 2.0, 4.0, and 8.0 mmol/L, with the control group receiving 0 mmol/L Al treatment. Eight physiological indices were measured, namely, (**A**) SOD (superoxide dismutase) activity; (**B**) POD (peroxidase) activity; (**C**) CAT (catalase) activity; (**D**) APX (ascorbate peroxidase) activity; (**E**) content of soluble protein; (**F**) content of soluble sugar; (**G**) proline content, and (**H**) malondialdehyde (MDA) content. The results were represented by the average value and SD (standard deviation) of three repeated biological experiments. The Waller–Duncan multiple comparison test and single-factor ANOVA were adopted to make a comparison for significant differences between the control group and treatment group with each concentration of Al toxicity. Different letters on the bar chart indicated significant differences between data (*p* < 0.05).

**Figure 5 plants-13-00325-f005:**
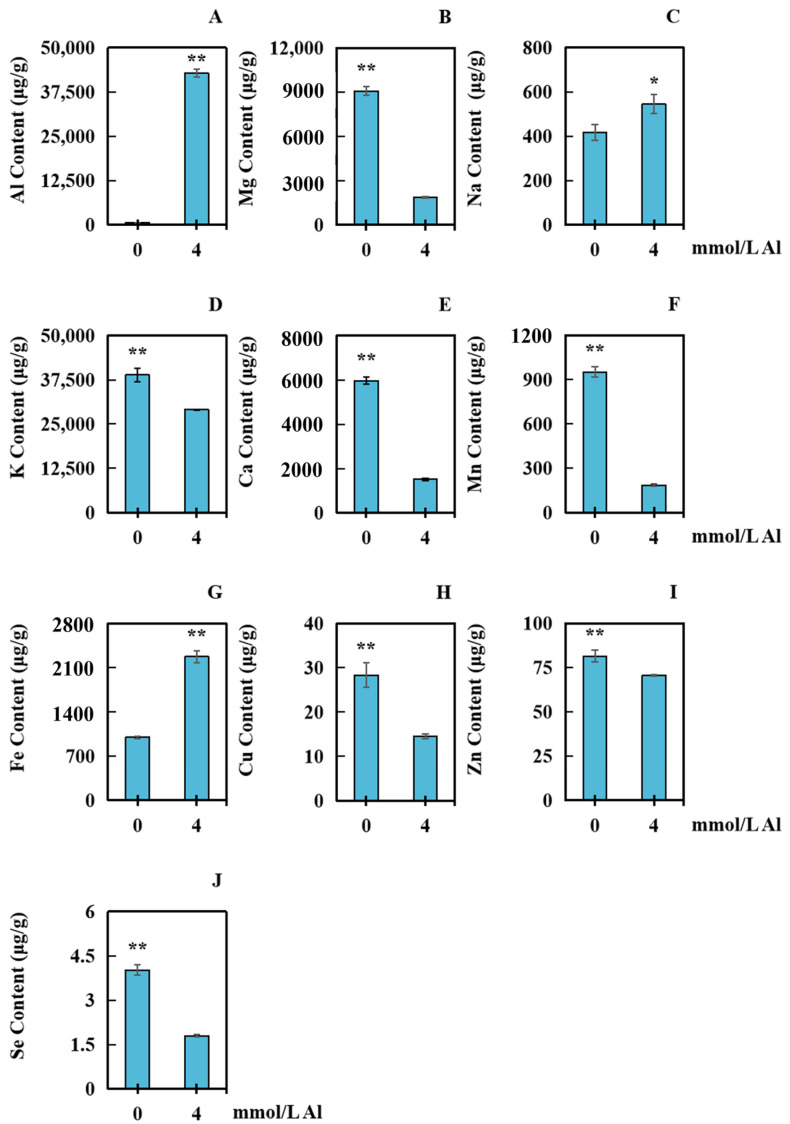
Accumulation of 10 elements in roots of peanuts subjected to aluminum poisoning. The samples were taken and dried after a 20-day treatment with 4.0 mmol/L exogenous aluminum, with the control group receiving 0 mmol/L aluminum treatment. The concentrations of the 10 elements were determined: (**A**) Na; (**B**) Mg; (**C**) Al; (**D**) K; (**E**) Ca; (**F**) Mn; (**G**) Fe; (**H**) Cu; (**I**) Zn; and (**J**) Se. The results were represented by the average value and SD (standard deviation) of three repeated biological experiments. The significant difference between the control group and the concentration of aluminum toxicity stress group was determined using an independent sample T test. Asterisks on the bar chart indicated significant (* *p* < 0.05) or very significant differences (** *p* < 0.01) between data.

**Figure 6 plants-13-00325-f006:**
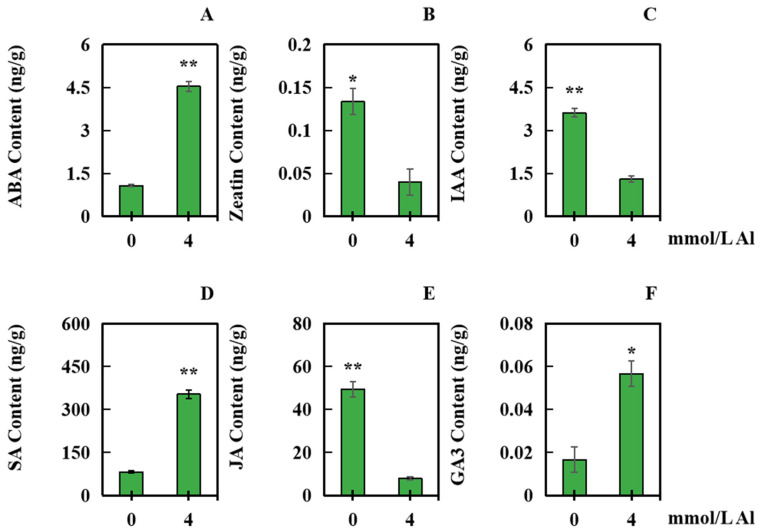
Changes in the levels of six hormones in roots of peanuts subjected to aluminum poisoning. The samples were taken after a 20-day treatment with 4.0 mmol/L exogenous aluminum, with the control group receiving 0 mmol/L aluminum treatment. The contents of the six hormones were determined: (**A**) ABA, (**B**) Zeatin, (**C**) IAA, (**D**) SA, (**E**) JA, and (**F**) GA_3_. The results were represented by the average value and SD (standard deviation) of three repeated biological experiments. The significant difference between the control group and the concentration of aluminum toxicity stress group was determined using an independent sample T test. Asterisks on the bar chart indicated significant (* *p* < 0.05) or very significant differences (** *p* < 0.01) between data.

**Figure 7 plants-13-00325-f007:**
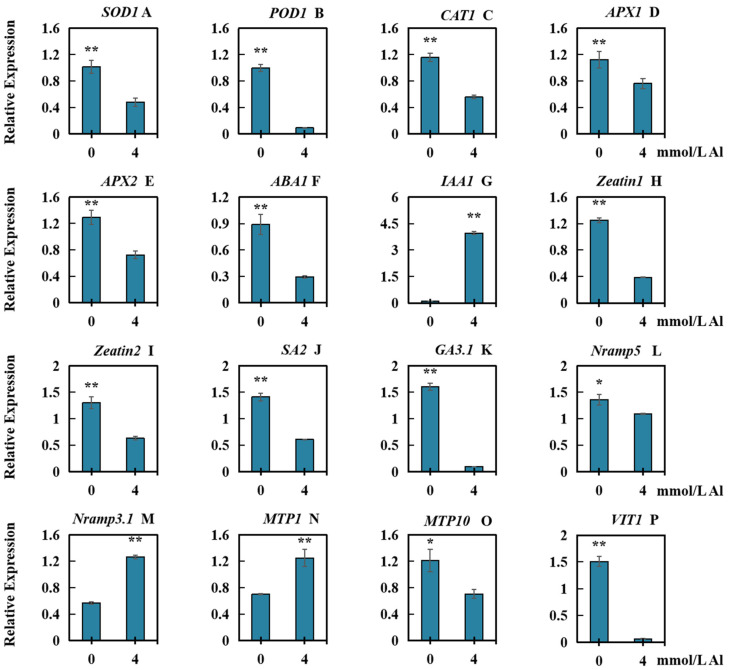
Relative expression of 15 genes in roots of peanuts under Al toxicity stress. Treatment with 0 mmol/L Al was regarded as the control, and treatment with 4.0 mmol/L exogenous Al was applied for 20 days. (**A**) *SOD1* (*superoxide dismutase 1*); (**B**) *POD1* (*peroxidase 1*); (**C**) *CAT1* (*catalase 1*); (**D**) *APX1* (*ascorbate peroxidase 1*); (**E**) *APX2* (*ascorbate peroxidase 1*); (**F**) *ABA1* (*abscisic acid 1*); (**G**) *IAA1* (*indoleacetic acid 1*); (**H**) *Zeatin1*; (**I**) *Zeatin2*; (**J**) *SA2* (*salicylic acid 2*); (**K**) *GA3.1* (*gibberellic acid 3.1*); (**L**) *Nramp5* (*natural resistant associated macrophage protein 5*); (**M**) *Nramp3.1* (*natural resistant associated macrophage protein 3.1*); (**N**) *MTP1* (*metal tolerance protein 1*); (**O**) *MTP10* (*metal tolerance protein 10*), (**P**) *VIT1* (*vacuole ion transporter 1*). The results were represented by the average value and SD (standard deviation) of three repeated biological experiments. The significant difference between the control group and the concentration of aluminum toxicity stress group was determined using an independent sample *t* test. Asterisks on the bar chart indicated significant (* *p* < 0.05) or very significant differences (** *p* < 0.01) between data.

**Figure 8 plants-13-00325-f008:**
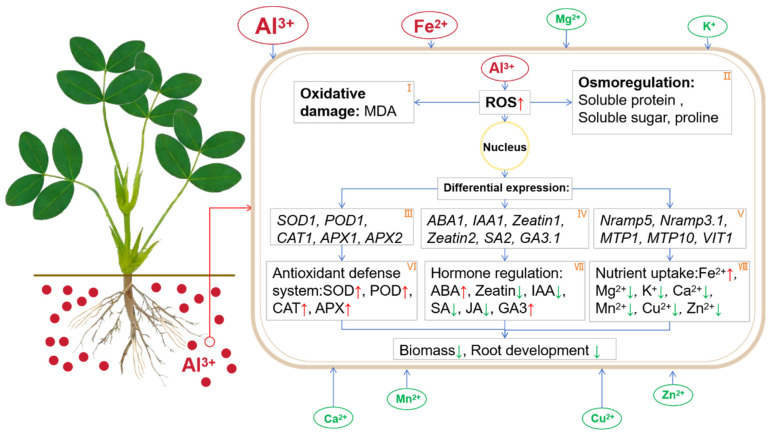
Regulatory pathway of aluminum poisoning in peanut roots. (**I**) Oxidative damage; (**II**) osmotic control; (**III**) antioxidase-related genes; (**IV**) hormone-related genes; (**V**) genes related to ion transport; (**VI**) antioxidant enzyme activity; (**VII**) hormone content; (**VIII**) absorption of mineral nutrients. Small red arrows indicated the increased content of substances or the increased activity of enzymes. The small green arrows indicated decreased enzyme activity, reduced substance content, or inhibition.

## Data Availability

Data is contained within the article and Appendix A.

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
