# Peer review of "Physiological Mechanism through Which Al Toxicity Inhibits Peanut Root Growth"

_plants, 2024, doi:10.3390/plants13020325_

Round 1

Reviewer 1 Report

Comments and Suggestions for Authors

The approach used in the study to evaluate physiological mechanism in peanut root under Al toxicity conditions is solid and well-documented. However a minor review might improve the research’s openness.

Please arrange the chapters of the article in the right order: insert "materials and methods" immediately after the introduction, followed by the "results".

Suggestions for Authors

-        Give more information on the peanut variety: is it the most cultivated in China? Is it sensitive to Al toxicity?

-        How many plants for treatments?

-        Was the nutrient solution aerated? How was it?

-        What was the pH of the solution?

-        Sometimes you talk about plants, other times about peanut roots, clarify this better!

-        It is better to speak about “control group” than “normal group”

-        Boron (B) deficiency and aluminum (Al) toxicity are two major constraints on plants grown in acidic soils. You have evaluated the influence of Al poisoning on several elements but not B, why?

-        In the "Data Analysis" subparagraph it specifies the tests used better since you are talking about Student's tests and in the captions of the tables we see the Waller-Duncan test.

-        In "Materials and methods" you talk about 6 treatments including the control group and in the “Results” the differences between these 6 treatments are reported only for the root biomass and development, and physiological response indices, but not for the accumulation of elements, hormones and gene transcription. For these results only use the 4mM concentration. Why? Why did you choose only this concentration versus control? For example, you have to correct the text on line 211 where you indicate "various Al concentrations" but in reality it is only one (4 mM).

-        Some considerations resulted from previous research by other Chinese authors not reported in this work (see references below as a suggestion).

Dong Xiao, Xia Li, Yun-Yi Zhou, Li Wei, Chanthaphoone Keovongkod, Huyi He, Jie Zhan, Ai-Qin Wang, Long-Fei He. Transcriptome analysis reveals significant difference in gene expression and pathways between two peanut cultivars under Al stress. Gene 781 (2021) 145535. https://doi.org/10.1016/j.gene.2021.145535

Wenjing Huang, Xudong Yang, Shaochang Yao, Thet LwinOo, Huyi He, Aiqin Wang, Chuangzhen Li, Longfei He. Reactive oxygen species burst induced by aluminum stress triggers mitochondria-dependent programmed cell death in peanut root tip cells. Plant Physiology and Biochemistry 82 (2014) 76e84. http://dx.doi.org/10.1016/j.plaphy.2014.03.037

Lei Yan, Muhammad Riaz, Shuang Li, Jin Cheng, Cuncang Jiang. Harnessing the power of exogenous factors to enhance plant resistance to aluminum toxicity; a critical review. Plant Physiology and Biochemistry 203 (2023) 108064. https://doi.org/10.1016/j.plaphy.2023.108064

-        Please, put the doi number in all references

Author Response

My responses

Reviewer 1

  1. Please arrange the chapters of the article in the right order: insert "materials and methods" immediately after the introduction, followed by the "results".

My response: The chapters of the article in the order: 1. Introduction, 2. Results, 3. Discussion, 4. Materials and Methods, 5. Conclusions. This is strictly sorted according to the format requirements of the journal of Plants. Two papers from Plants published in 2024 were randomly selected, and they were formatted in this format: 1. Introduction, 2. Results, 3. Discussion, 4. Materials and Methods, 5. Conclusions.

Yang, S.; Ning, Y.; Li, H.; Zhu, Y. Effects of Priestia aryabhattai on Phosphorus Fraction and Implications for Ecoremediating Cd-Contaminated Farmland with Plant–Microbe Technology. Plants 2024, 13, 268. https://doi.org/10.3390/plants13020268

Wang, Z.; Sanusi, I.A.; Wang, J.; Ye, X.; Kana, E.G.; Olaniran, A.O. Biogas Slurry Significantly Improved Degraded Farmland Soil Quality and Promoted Capsicum spp. Production. Plants 2024, 13, 265. https://doi.org/10.3390/plants13020265

  1. Give more information on the peanut variety: is it the most cultivated in China? Is it sensitive to Al toxicity?

My response: The peanut variety named Zhanyou 62 is not the most cultivated peanut variety in China, but it is widely cultivated in the peanut producing areas of southern China. Prior to our study, no studies have shown that Zhanyou 62 is resistant to aluminum or sensitive to aluminum.

  1. How many plants for treatments?

My response: Each treatments has 18-24 plants.

  1. Was the nutrient solution aerated? How was it?

My response: Aerating pumps for plant hydroponics were used to aerate gas in nutrient solution for 20 minutes every hour.

  1. What was the pH of the solution?

My response: The pH of the solution was at 4.5.

  1. Sometimes you talk about plants, other times about peanut roots, clarify this better!

My response: Line 57, 75, 78. This is good advice. I have modified them.

  1. It is better to speak about “control group” than “normal group”

My response: Line 125, 129, 152, 162, 191, 198, 206, 214, 217, 225, 233, 236, 262, 356, 526. Thank you for your suggestion. I have changed “normal group” to “control group”.

  1. Boron (B) deficiency and aluminum (Al) toxicity are two major constraints on plants grown in acidic soils. You have evaluated the influence of Al poisoning on several elements but not B, why?

My response: Thank you for your very good advice. In this study, we only focused on the relationship between the content changes of these ten elements (Na, Mg, Al, K, Ca, Mn, Fe, Cu, Zn, and Se) and aluminum toxicity stress. We'll focus on boron (B) deficiency and aluminum (Al) toxicity in our future research.

  1. In the "Data Analysis" subparagraph it specifies the tests used better since you are talking about Student's tests and in the captions of the tables we see the Waller-Duncan test.

My response: In this study, Student's tests were used to compare the significance of differences between the two groups of data. The Waller-Duncan test was used to compare the significance of differences across multiple groups of data. I have modified them. (Line 607-608)

  1. In "Materials and methods" you talk about 6 treatments including the control group and in the “Results” the differences between these 6 treatments are reported only for the root biomass and development, and physiological response indices, but not for the accumulation of elements, hormones and gene transcription. For these results only use the 4mM concentration. Why? Why did you choose only this concentration versus control? For example, you have to correct the text on line 211 where you indicate "various Al concentrations" but in reality it is only one (4 mM).

My response: Line 223-224. I've modified it. “The samples were taken under normal aluminum conditions and at 4 mmol/L Al.”

We believe that the Al concentration of 4 mM is more representative in the treatment of this study. Peanuts growing at this concentration have obvious aluminum toxicity stress phenotype, and the damage is not as extreme as 8 mM, so it has high representativeness and research value. Therefore, the concentration of 4 mM Al (aluminum toxicity concentration) was chosen as the representative to study the element accumulation, hormone content and gene expression of peanuts.

  1. Some considerations resulted from previous research by other Chinese authors not reported in this work (see references below as a suggestion).

Dong Xiao, Xia Li, Yun-Yi Zhou, Li Wei, Chanthaphoone Keovongkod, Huyi He, Jie Zhan, Ai-Qin Wang, Long-Fei He. Transcriptome analysis reveals significant difference in gene expression and pathways between two peanut cultivars under Al stress. Gene 781 (2021) 145535. https://doi.org/10.1016/j.gene.2021.145535

Wenjing Huang, Xudong Yang, Shaochang Yao, Thet LwinOo, Huyi He, Aiqin Wang, Chuangzhen Li, Longfei He. Reactive oxygen species burst induced by aluminum stress triggers mitochondria-dependent programmed cell death in peanut root tip cells. Plant Physiology and Biochemistry 82 (2014) 76e84. http://dx.doi.org/10.1016/j.plaphy.2014.03.037

Lei Yan, Muhammad Riaz, Shuang Li, Jin Cheng, Cuncang Jiang. Harnessing the power of exogenous factors to enhance plant resistance to aluminum toxicity; a critical review. Plant Physiology and Biochemistry 203 (2023) 108064. https://doi.org/10.1016/j.plaphy.2023.108064

My response: I have cited these three literatures in the paper.

L42, 62-64: 7. Huang, W.J.; Yang, X.D.; Yao, S.C.; LwinOo, T.; He, H.Y.; Wang, A.Q.; Li, C.Z.; He, L.F. Reactive oxygen species burst induced by aluminum stress triggers mitochondria-dependent programmed cell death in peanut root tip cells. Plant Physiol. Bioch. 2014, 82, 76–84.

L67, 78-80: 13. Yan, L.; Riaz, M.; Li, S.; Cheng, J.; Jiang, C.C. Harnessing the power of exogenous factors to enhance plant resistance to aluminum toxicity; a critical review. Plant Physiol. Bioch. 2023, 203, 108064.

L69, 73-74: 14. Xiao, D.; Li, X.; Zhou, Y.Y.; Wei, L.; Keovongkod, C.; He, H.Y.; Zhan, J.; Wang, A.Q.; He, L.F. Transcriptome analysis reveals significant difference in gene expression and pathways between two peanut cultivars under Al stress. Gene 2021, 781, 145535.

  1. Please, put the doi number in all references

My response: This is good advice. We strictly follow the format requirements of the journal. However, journal references in Plants are not required to include the doi number. I randomly selected four newly published papers in 2024, none of which provided doi number in their references.

Yang, S.; Ning, Y.; Li, H.; Zhu, Y. Effects of Priestia aryabhattai on Phosphorus Fraction and Implications for Ecoremediating Cd-Contaminated Farmland with Plant–Microbe Technology. Plants 2024, 13, 268. https://doi.org/10.3390/plants13020268

Wang, Z.; Sanusi, I.A.; Wang, J.; Ye, X.; Kana, E.G.; Olaniran, A.O. Biogas Slurry Significantly Improved Degraded Farmland Soil Quality and Promoted Capsicum spp. Production. Plants 2024, 13, 265. https://doi.org/10.3390/plants13020265

Quezada-D’Angelo, T.; San Martín, J.; Ruiz, B.; Oyarzúa, P.; Vargas, M.; Fischer, S.; Cortés, P.; Astete, P.; Moya-Elizondo, E. Use of Pseudomonas protegens to Control Root Rot Disease Caused by Boeremia exigua var. exigua in Industrial Chicory (Cichorium intybus var. sativum Bisch.). Plants 2024, 13, 263. https://doi.org/10.3390/plants13020263

Gao, Y.; Sun, C.; Ramos, T.B.; Tan, J.; Oliveira, A.R.; Huang, Q.; Huang, G.; Xu, X. Global Sensitivity Analysis of the Advanced ORYZA-N Model with Different Rice Types and Irrigation Regimes. Plants 2024, 13, 262. https://doi.org/10.3390/plants13020262

Reviewer 2 Report

Comments and Suggestions for Authors

The study etitled "Physiological Mechanism through Which Al Toxicity Inhibits Peanut Root Growth", is overal well organized, the used methods are new and the obtained results are useful for the researchers in the field. There are some isues that author must address:

a. the exact aim of the study is not well presented, please add it in the abstract and in the introduction part.

b. what was the reason of using only 4.0 mmol/L Al for ion content, hormones and PCR and not other concentration.

c. in the caption of the charts please specify the meaning of a, b, c and d as was mentioned for other marks (e.g. * and **).

Author Response

Reviewer 2

  1. the exact aim of the study is not well presented, please add it in the abstract and in the introduction part.

My response: Thank you for your advice. I've revised those paragraphs.

L26-28: The purpose of this study was to explore the physiological response mechanism of peanut roots subjected to aluminum toxicity stress, and the findings of this research will provide a basis for cultivating Al-resistant peanut new varieties.

L94-98: The goal of this study was to investigate the physiological response mechanism of peanut roots to aluminum toxicity stress, and the findings would serve as a foundation for the development of novel Al-resistant peanut types.

  1. what was the reason of using only 4.0 mmol/L Al for ion content, hormones and PCR and not other concentration.

My response: We believe that the Al concentration of 4 mM is more representative in the treatment of this study. Peanuts growing at this concentration have obvious phenotype of aluminum toxicity stress, and the damage is not as extreme as 8 mM, so it has high representativeness and research value. Therefore, the concentration of 4 mM Al (aluminum toxicity concentration) was chosen as the representative to study the element accumulation, hormone content and gene expression of peanuts.

  1. in the caption of the charts please specify the meaning of a, b, c and d as was mentioned for other marks (e.g. * and **).

My response: Thank you for your advice. I have added related instructions in the picture notes.

L130, 153, 198: Different letters on the bar chart indicated significant differences between data (p < 0.05).

L219, 238, 264: Asterisks on the bar chart indicated significant (* P<0.05) or very significant differences (** P<0.01) between data.
